# Robust Online Correlation Clustering

**Silvio Lattanzi**
Google Research
silviol@google.com

**Benjamin Moseley**
Carnegie Mellon University
moseleyb@andrew.cmu.edu

**Sergei Vassilvitskii**
Google Research
sergeiv@google.com

**Yuyan Wang**
Carnegie Mellon University
yuyanw@andrew.cmu.edu

**Rudy Zhou**
Carnegie Mellon University
rbz@andrew.cmu.edu

## Abstract

In correlation clustering we are given a set of points along with recommendations whether each pair of points should be placed in the same cluster or into separate clusters. The goal is to cluster the points to minimize disagreements from the recommendations. We study the correlation clustering problem in the online setting, where points arrive one at a time, and upon arrival the algorithm must make an irrevocable cluster assignment decision. While the online version is natural, there is a simple lower bound that rules out any algorithm with a non-trivial competitive ratio. In this work we go beyond worst case analysis, and show that the celebrated Pivot algorithm performs well when given access to a small number of random samples from the input. Moreover, we prove that Pivot is robust to additional adversarial perturbations of the sample set in this setting. We conclude with an empirical analysis validating our theoretical findings.

## 1  Introduction

Clustering is a central unsupervised learning problem, which comes in many flavors and variants. For instance we may consider different objectives for the clustering: $k$-clustering like $k$-means or $k$-median, spectral clustering, or correlation clustering. Or we may try to perform the clustering in different computational models—batch, distributed, parallel, etc. A setting that has received a lot of interest in recent years is the online paradigm, see for instance, Zhang et al. [2004], Vidal [2006], Lattanzi and Vassilvitskii [2017], Cohen-Addad et al. [2019], Guo et al. [2020, 2021]. In online clustering elements arrive one by one and they are irrevocably assigned to clusters at arrival time. A key property of this setting is that the resulting clustering is stable. This is particularly important in real-world systems where clusters are served directly to users or used in downstream machine learning tasks, and so modifying the clustering has a high cost.

Clustering problems like $k$-median and $k$-means are well studied in the online setting, see for instance the groundbreaking work of Meyerson [2001] and Liberty et al. [2016]. This paper considers the online correlation clustering problem, which is less understood. Correlation clustering was introduced by Bansal et al. [2004]. In the problem we are given as input a weighted graph with $n$ nodes, where positive edges represent similarities and negative edges represent dissimilarities between endpoints. A partition of the nodes into clusters should minimize the sum of the negative edges contained in any cluster and the sum of positive edges between clusters. That is, minimizing the *disagreements* in the clustering. Correlation clustering has many applications, for instance in finding clustering ensembles[Bonchi et al., 2013], duplicate detection [Arasu et al., 2009], community mining [Chen et al., 2012], disambiguation tasks [Kalashnikov et al., 2008], and automated labelling [Agrawal et al., 2009, Chakrabarti et al., 2008].

35th Conference on Neural Information Processing Systems (NeurIPS 2021).

The problem is known to be NP-hard and admits approximation algorithms. For the most studied version of the problem, where the weights are restricted to be in $\{-1, +1\}$, and a 2.06 approximation algorithm was shown by Chawla et al. [2015]. When, in addition, the number of clusters is upper-bounded by $k$, a polynomial time approximation scheme was shown to exist by Giotis and Guruswami [2005]. For arbitrary weights a $O(\log n)$ approximation is the best known [Demaine et al., 2006].

We consider the online correlation clustering problems when edges have weights in $\{-1, +1\}$. In the online setting, nodes arrive one by one and reveal their edges to nodes that have arrived previously. The algorithm needs to make an irrevocable cluster assignment when a node arrives. The most popular form of analysis of online algorithms is *competitive analysis* where the performance of the online algorithm is compared to the best (in hindsight) optimum solution.

It is *impossible* to design an online algorithm for correlation clustering that has a good competitive ratio. Consider the following example. Suppose that the first two nodes to arrive are connected by a positive edge. The algorithm has to commit to either placing them in the same cluster, or in different clusters. If it places them in the same cluster then the adversary places the remaining $n - 2$ nodes as follows. The nodes are split evenly between two complete cliques with all positive edges. The two initial nodes are placed in different cliques and there are no positive edges between cliques except the single edge between the two initially arriving nodes. The optimal solution in hindsight makes 1 disagreement by placing the cliques into two separate clusters, while the algorithm makes at least $\Theta(n)$ disagreements.

On the other hand if the algorithm places the first two nodes into different clusters then the adversary chooses the remaining instance to be a single clique of positive edges containing the two nodes. The competitive ratio of the algorithm will be unbounded as the optimal has cost 0 and the algorithm has a positive objective value of at least one.

This lower bound example (first observed in Mathieu et al. [2010]) rules out finding a good algorithm via competitive analysis, since *every* algorithm will have a worst case competitive ratio of $\Omega(n)$. However, in practice, instances are typically far from worst case. To rigorously study these situations requires going beyond worst case analysis. For a rich body of work on this topic see the recent text by Roughgarden [2020]. This line of work considers broadly applicable models that avoid pathological worst-case instances that are unlikely to occur in practice.

In practice, for online clustering problems, it is common to have some information about nodes that will arrive. For example when clustering temporal data we have access to last month's data and this data gives some insights on tomorrow's data. Motivated by this, it is natural to study a model where one has access to few samples of the input data representing the past data that has been observed. Of course, if the sample is completely adversarial the problem has not become any easier. To overcome this limitation we follow previous work [Korula et al., 2018, Lykouris et al., 2018] and consider a setting where the algorithm has access to a random sample of data.

The model considered is as follows. Suppose that for any (potentially adversarially chosen) input, the algorithm is given a small number of random samples from the input before the rest is revealed online. Can we prove that it is possible to design online algorithms that are near optimal? If possible, can the model be strengthened such that some of the sample is randomized while some is adversarial and we can design robust online algorithms for this more challenging setting? We answer positively to both questions in this work.

**Our Contributions.** We introduce the *semi-online* model for correlation clustering. In this model elements arrive online in adversarial order but beforehand a small random fraction of nodes are revealed. This setting captures situations where some part of the online input is available beforehand, a common case when analyzing temporal data. A similar semi-online model for bipartite matching is introduced by Kumar et al. [2019], where a part of the input graph is predictable and known offline, while the remainder is unknown and arrives online.

We prove that the well-know Pivot algorithm [Ailon et al., 2008] obtains a constant fraction approximation in the online setting when a small fraction of points selected at random is revealed beforehand. The key insight behind our proof is that the Pivot algorithm can use a small random sample to "sparsify" the input instance so that it is easily clusterable.

Furthermore, we show that even if a fraction of the sample is adversarially corrupted the Pivot algorithm can still recover a constant approximation. In other words, the approach is provably robust.

Finally we analyze our finding empirically and verify that the theory is predictive of empirical results. We show that even on "adversarial" inputs the Pivot algorithm obtains good performance if it has access to a small sample of the data. Without the sample, Pivot performs poorly. We further show similar behaviour can be observed when the sample is learned from historical data. These results demonstrate the connection between the theoretical model and practice.

**Additional related work.** The online correlation clustering problem has been studied by Mathieu et al. [2010] , where they give a $O(n)$-competitive algorithm. However, their algorithm is allowed to re-assign points by merging clusters. Ailon et al. [2008] show that the Pivot algorithm is 3-competitive when vertices arrive in random order.

Our setting is also closely related to the algorithm with machine learning advice setting Lykouris and Vassilvitskii [2018]. In this context, Ailon et al. [2018], Saha and Subramanian [2019] study the correlation clustering problem and show that it is possible to obtain better performance guarantees using same-cluster queries.

## 2 Semi-Online Model for Correlation Clustering

In this paper, we consider the min-disagreement *correlation clustering* problem. We first introduce the offline version of the problem.

**Definition 1** (Correlation Clustering). *The input is a complete signed graph $G = (V, E)$ on $n$ vertices such that the edges are partitioned into positive and negative edges $E = E^+ \cup E^-$, respectively.*

*The goal is to partition vertices into clusters, e.g. $V = C_1 \cup \cdots \cup C_k$, where $k$ is arbitrary, to minimize the number of disagreements: number of positive edges with endpoints in different clusters plus the number of negative edges within each cluster. We can write this objective as: $\sum_{i<j} |\{uv \in E^+ \mid u \in C_i, v \in C_j\}| + \sum_{i=1}^{k} |\{uv \in E^- \mid u, v \in C_i\}|$.*

In the *online* model, the vertices of the graph arrive one-by-one online. The algorithm has to maintain a partition of all arrived vertices into clusters at all times. Upon each vertex arrival, its edges (and their signs) to all previous arrivals are revealed. Then the algorithm must make an irrevocable cluster assignment for this arrival. We recall the lower bound in this setting due to Mathieu et al. [2010].

**Lemma 1.** *[Mathieu et al., 2010] For any $n$, there exists an instance of online correlation clustering where vertices arrive in adversarial order such that every algorithm has competitive ratio $\Omega(n)$.*

**Semi-Online Model.** As discussed, it is impossible to obtain any positive result in the online model. Thus, we introduce the semi-online model for correlation clustering.

**Definition 2** (Semi-Online Correlation Clustering). *Let $G = (V, E)$ be a complete signed graph. In semi-online correlation clustering, there are two phases: an offline and online phase.*

*First, in the offline phase, our algorithm is given the induced subgraph $G[S]$ for some vertex set $S \subset V$. Then, in the online phase, the remainder of the vertices $V \setminus S$ arrive online. Upon each online arrival, the algorithm must allocate the vertex to a cluster, either starting a new singleton cluster, or assigning it to a previously defined one.*

*The goal is to maintain a clustering of the vertices throughout to minimize the number of disagreements.*

We will consider different models for how the offline vertices $S$ are chosen. We note that if $S$ is chosen adversarially, then the adversary can choose $S$ to be a graph with only negative edges, and $V \setminus S$ to be a lower bound instance of size $n - |S|$. This gives a lower bound of $\Omega(n - |S|)$. Thus, we focus on models where $S$ is, at least, partially chosen randomly.

The algorithm we analyze for the semi-online correlation clustering is the natural adaptation of the *Pivot algorithm* developed by Ailon et al. [2008]. We describe it first.

**Pivot algorithm**

1. Initially, all vertices are *unclustered*.
2. Consider vertices $v \in V$ in some order.
3. If $v$ is currently unclustered, then mark $v$ as a *pivot*, and make a new cluster consisting of $v$ and its remaining unclustered, positive neighbors. Thus, the new cluster is $\{v\} \cup \{u \in V \mid uv \in E^+, u \text{ unclustered}\}$.

4. Repeat until all vertices are clustered, and output the final clustering.

Pivot's performance critically depends on the ordering of the vertices. If the algorithm considers vertices in random order, then Pivot is a 3-approximation in expectation [Ailon et al., 2008].

Pivot can be adapted to the online setting by considering the vertices in their arrival order. As we noted, this algorithm will perform poorly on adversarially ordered sequences; moreover as we will see it has poor empirical performance as well. In the semi-online model, we can take advantage of the offline phase to find better clusters for the online arriving vertices via the pivots computed in the offline phase.

**Semi-Online Pivot**

1. In the offline phase, randomly order the vertices in $S$ and then using the random ordering run the Pivot algorithm on $G[S]$.
2. In the online phase, we continue running the Pivot algorithm on the remainder of $G$ given the pivots and resulting clustering on $G[S]$. Note that we consider vertices in the order they arrive online.

Note that the pivots chosen in the offline phase can potentially cluster vertices in the online phase. In particular, if vertex $v$ arrives in the online phase, and $v$ has a positive edge to a pivot $u$ chosen in the offline phase, then $v$ joins $u$'s cluster. We will show that if $S$ is a $\epsilon n$-sized random sample of the input, then Pivot is $O(\frac{1}{\epsilon})$-competitive. Thus a small random sample is sufficient to circumvent the strong lower bounds on any online algorithm. We also show that this style of algorithm is robust, even when $S$ is not fully random, semi-online Pivot is guaranteed to perform well.

## 3 Warm-Up: Pivoting Using a Random Sample

In this section, we consider the case where $S$ is a random sample from $V$ of size $\epsilon n$ for $\epsilon \in (0, 1)$. We call this the $\epsilon$-*random sample model*. The analysis of this setting will illustrate the main technical ideas. Later we generalize to the case where the random sample has a fraction of adversarially chosen corruptions. In this section, we prove the following.

**Theorem 1.** *Fix an* $\epsilon \in (0, 1)$, *and let* $S$ *have* $\epsilon n$ *samples chosen uniformly at random. Then semi-online Pivot is* $O(\frac{1}{\epsilon})$*-competitive in expectation.*

We also complement our upper bound with a matching lower bound.

**Theorem 2.** *For any* $\epsilon \in (0, 1)$, *every algorithm in the* $\epsilon$-*random sample model has competitive ratio* $\Omega(\frac{1}{\epsilon})$.

*Proof.* To prove the lower bounds, it suffices to consider $\epsilon < \frac{1}{2}$. Fix a sufficiently large $n$ such that $\frac{1}{\epsilon} + \epsilon n \leq n$. We define a graph on $n$ nodes. All edges are negative except those on a set $L$ of $\frac{1}{\epsilon}$ vertices. The edges on $L$ form a lower bound instance of size $\frac{1}{\epsilon}$ guaranteed by Lemma 1.

The probability that the random sample $S$ contains no vertex from $L$ is: $\mathbb{P}(S \cap L = \emptyset) = (1 - \frac{1/\epsilon}{n}) \dots (1 - \frac{1/\epsilon}{n - \epsilon n + 1}) \geq (1 - \frac{1/\epsilon}{n - \epsilon n})^{\epsilon n} = \Omega(1)$. Further, conditioned on this event, the cost of any algorithm is $\Omega(\frac{1}{\epsilon})$-competitive, because $L$ arrives in adversarial order. $\qquad\square$

We now prove the upper bound. Our proof will proceed in two parts. First, we bound the cost of semi-online Pivot on the random sample $S$. Recall that these vertices arrive in random order, so we can leverage the analysis of the original Pivot algorithm.

Second, we bound the cost of $V \setminus S$. The main difficulty here is that $V \setminus S$ arrives in adversarial order. To overcome the $\Omega(n)$ lower bound, we first prove a more refined lower bound on the optimal solution in terms of the positive degrees (number of positive edges incident on a vertex) in the input graph. Then we show that running semi-random Pivot on $S$ sparsifies the remaining graph on $V \setminus S$. That is, there are few positive edges on the nodes no adjacent to a pivot in the sample. Finally, we relate the cost of Pivot to the positive degrees to complete the proof.

### 3.1  Pivot Preliminaries

Throughout this section, let OPT denote the number of disagreements made by the optimal offline algorithm on $G$ and ALG denote the cost of semi-online Pivot. Note that OPT is a fixed quantity, while ALG depends on the random choice of $S$ and its order.

We begin with the concept of *bad triangles*, which we use to lower bound OPT and upper bound ALG.

**Definition 3** (Bad Triangle). *A bad triangle in $G$ is a triple of vertices, $t = ijk$ such that two edges among them are positive, and the remaining edge is negative. We let $T$ denote the set of bad triangles in $G$.*

By case analysis one can check that any clustering must make at least one disagreement in a bad triangle. Similarly, the Pivot algorithm makes a disagreement on edge $ij$ if and only if there exists a bad triangle $ijk$ of unclustered vertices such that $k$ is chosen as pivot. This motivates the definition for any bad triangle $t$ of the event $A_t = \{$one vertex of $t$ is chosen as a pivot while all three are unclustered$\}$. By linearity of expectation, we can write our expected cost as:

$$\mathbb{E}[\text{ALG}] = \sum_{t \in T} \mathbb{P}(A_t). \tag{1}$$

We emphasize that this expression holds for any distribution of arrivals. The only difference is the probabilities of the $A_t$'s. We will re-use the same notations and expression when we consider a random sample with corruptions later. However, for the remainder of this section, all probabilities are with respect to the $\epsilon$-random sample model unless otherwise noted.

To formalize the two main steps in our proof sketch, we decompose the above quantity into two parts: one for $S$ and one for $V \setminus S$. To this end, we define for each bad triangle $t$ the events $A_t^S = \{$one vertex of $t$ is chosen as pivot in $S$ while all three are unclustered$\}$ and $A_t^{V \setminus S} = \{$one vertex of $t$ is chosen as pivot in $V \setminus S$ while all three are unclustered$\}$.

We can re-write: $\mathbb{E}[\text{ALG}] = cost(S) + cost(V \setminus S)$, where $cost(S) = \sum_{t \in T} \mathbb{P}(A_t^S)$ and $cost(V \setminus S) = \sum_{t \in T} \mathbb{P}(A_t^{V \setminus S})$. We will analyze each of these two terms separately. In particular, we first show that $cost(S) = O(\text{OPT})$ by relating to Pivot in random order. Then we show $cost(V \setminus S) = O(\frac{1}{\epsilon})\text{OPT}$ by relating to the positive degrees. Combining these two bounds with the above expression for $\mathbb{E}[\text{ALG}]$ completes the proof of Theorem 1.

### 3.2  Bounding Cost of $S$

We show $cost(S) = O(\text{OPT})$. Because the vertices of $S$ are a random subset of $V$ and we can analyze them in random order, the first $\epsilon n$ arrivals of the $\epsilon$-random sample model are distributed identically to the first $\epsilon n$ arrivals in the model where *all* vertices arrive in random order. We rely on the next theorem from Ailon et al. [2008], which bounds the cost of Pivot when vertices arrive in random order.

**Theorem 3.** *[Ailon et al., 2008] If vertices arrive in random order, then Pivot is 3-competitive in expectation.*

Now we use that Equation (1) holds for both the $\epsilon$-random sample model and the random order model. We denote the former distribution with a subscript $S$ and the latter with a subscript $R$. Let RAND denote the cost of running Pivot in the random order model. Then applying Equation (1) and Theorem 3, we have: $cost(S) = \sum_{t \in T} \mathbb{P}_S(A_t^S) = \sum_{t \in T} \mathbb{P}_R(A_t^S) \leq \sum_{t \in T} \mathbb{P}_R(A_t) = \mathbb{E}_R[\text{RAND}] \leq 3\text{OPT}$.

The second equality follows from the observation that the first $\epsilon n$ arrivals in the $\epsilon$-random sample model are distributed identically to the first $\epsilon n$ arrivals in the random order model, and the event $A_t^S$ does not depend on the arrival order of $V \setminus S$. Thus for any bad triangle $t$, $\mathbb{P}_S(A_t^S) = \mathbb{P}_R(A_t^S)$. To understand the latter probability, in the random order model we use the convention that the ordered set $S$ is the first $\epsilon n$ arrivals.

### 3.3  Bounding Cost of $V \setminus S$

We show $cost(V \setminus S) = O(\frac{1}{\epsilon})\text{OPT}$. We begin with a lower bound on OPT in terms of the positive degrees. Because every clustering must make at least one disagreement on each bad triangle, we can

interpret a clustering as covering all bad triangles using edges (which are the disagreements that this clustering makes.) Further, every bad triangle has two positive edges, so we show that the number of bad triangles that any edge can cover is proportional to the positive degrees of its endpoints.

**Lemma 2.** *Fix any clustering $C$ and let $E^*$ denote the set of edges that $C$ makes a disagreement on. Then $|T| \leq \sum_{ij \in E^*} d^+(i) + d^+(j)$, where $d^+(\cdot)$ denotes the positive degree of a vertex.*

*Proof.* $C$ must make at least one disagreement on each bad triangle. It follows, $C$ must cover each bad triangle with at least one edge in $E^*$. It suffices to show that each $ij \in E^*$ can be in at most $d^+(i) + d^+(j)$ bad triangles. To see this, we consider two cases. If $ij \in E^-$, then each bad triangle including $ij$ must also have two positive edges $ik$ and $jk$ for some $k$. There can be at most $\min(d^+(i), d^+(j))$ such $k$. Otherwise, $ij \in E^+$. Then each bad triangle including $ij$ must have one other positive edge – either $ik$ or $jk$ for some $k$. There can be at most $d^+(i) + d^+(j)$ such $k$. $\qquad \square$

To understand the utility of the above lemma, it is informative to upper bound each $d^+(i)$ by the max positive degree, say $d^*$. Then Lemma 2 gives $|T| = O(d^*)\text{OPT}$. Further, by Equation (1), the expected cost of Pivot (for *any* arrival distribution) is $\sum_{t \in T} \mathbb{P}(A_t) \leq |T|$. Combining these two inequalities gives that Pivot is a $O(d^*)$-approximation even in adversarial order. In the worst case, we have $d^* = \Omega(n)$ (as in the standard $\Omega(n)$ lower bound instance.)

To overcome this, note that some nodes in $V \setminus S$ are "pre-clustered" by running Pivot on $S$. This occurs when a vertex in $V \setminus S$ has a positive edge to a pivot in $S$. Bad triangles containing such vertices do not contribute to $cost(V \setminus S)$, so it suffices to consider the remaining subgraph of vertices in $V \setminus S$ that are not pre-clustered. We show that this random subgraph has small positive degrees in expectation, so applying Lemma 2 to this sparse random subgraph allows us to bound $cost(V \setminus S)$.

We can now show our key structural lemma about sparsification. The main idea of the proof is that if a vertex has high positive degree, then it is likely that one of its positive neighbors becomes a pivot in $S$. Due to space, the proof can be found in Section B.

**Lemma 3.** *For any $v \in V$, define the random variable $N_v$ to be $0$ if $v$ is clustered by running Pivot on $S$ (i.e. $v \in S$ or $v$ has a positive edge to a pivot in $S$) or $N_v = |\{unclustered\ positive\ neighbors\ of\ v\ in\ V \setminus S\}|$ otherwise. Then $\mathbb{E}N_v = O(\frac{1}{\epsilon})$.*

To finish our bound on $cost(V \setminus S)$, it remains to combine Lemma 2 with Lemma 3. To this end, we let $G'$ denote the random subgraph of $G$ induced by all vertices in $V \setminus S$ that are not clustered by the pivots chosen in $S$. The key properties of $G'$ are:

- Let $T'$ be the set of all bad triangles in $G'$. We have $\sum_{t \in T} 1_{A_t^{V \setminus S}} \leq |T'|$. To see this, note that if $A_t^{V \setminus S}$ occurs, then no vertex of $t$ is pre-clustered by $S$, so $t \in T'$.
- The positive degree in $G'$ of a vertex $v$ is exactly $N_v$ (as defined in Lemma 3.) Note that here we use the fact that if $v$ is clustered by running Pivot on $S$, $N_v = 0$ , since $v$ is not a vertex in $G'$.
- Let $E'$ denote the edge set of $G'$ and $E^*$ the disagreements made by OPT on the whole graph $G$. Recall that OPT is defined on $G$, so $E^*$ is invariant. Then OPT induces a clustering of $G'$ that makes disagreements $E' \cap E^*$.

Using the above two lemmas and three properties, we conclude: $cost(V \setminus S) \leq \mathbb{E}|T'| \leq \mathbb{E}[\sum_{ij \in E' \cap E^*} d_{G'}^+(i) + d_{G'}^+(j)] \leq \sum_{ij \in E^*} \mathbb{E}N_i + \mathbb{E}N_j = O(\frac{1}{\epsilon})\text{OPT}$. This completes the proof of Theorem 1.

## 4   Random Sample with Adversarial Corruptions

We now consider the case where $S$ is a random sample from $V$ with adversarial corruptions. Our goal is to show that even with adversarial additions to $S$, pivot has strong approximation guarantees. We assume that $S$ is generated by the following procedure, parameterized by $\epsilon \in (0, 1)$ and $\alpha \in (0, \epsilon)$:

**$\epsilon$-Random Sample with $\alpha$-Corruption**

1. Given a complete signed graph $G = (V, E)$, an adversary chooses a fixed set $A \subset V$ of $\alpha n$ vertices.
2. Then a uniform random subset $R$ of size $(\epsilon - \alpha)n$ is drawn from $V \setminus A$.

3. Our algorithm is given $S = R \cup A$ in the offline phase.

We emphasize the order of events: first the adversary chooses $A$, and then a random sample $R$ is drawn. In particular, the choice of $A$ does not depend on $R$. Also, our algorithm is given $S$ in the offline phase, but critically it is unaware of which vertices belong to $A$ or $R$.

We briefly discuss some slight variants of our model and their tractability. Sampling $(\epsilon - \alpha)n$ from $V \setminus A$ rather than $\epsilon n$ from $V$ itself is mainly for technical convenience; the latter model admits similar theoretical guarantees, because $R \setminus A$ is chose to uniform on $V \setminus A$ if $R$ is a uniform random sample of $V$. However, if we change the order of events, so first sample $R$, and then allow an adversary to remove/replace $\alpha n$ vertices from the sample, then the competitive ratio degrades to $\Omega(\alpha n)$. To see this, consider an input graph consisting of a lower-bound instance of size $\alpha n$ guaranteed by Lemma 1 and all other edges are negative. The adversary can guarantee that the lower-bound instance is always outside the sample and thus arrives in adversarial order.

The main result of this section is that the corruptions do not degrade the performance of semi-online pivot; in particular, the algorithm performs as if it had an uncorrupted sample of size $(\epsilon - \alpha)n$ (under the mild assumption that $\alpha$ is at most a constant fraction of $\epsilon$.) We defer the proof of the next theorem to Section A

**Theorem 4.** *For $\epsilon \in (0,1)$ and $\alpha \in (0, \epsilon)$, semi-online pivot in the $\epsilon$-random sample with $\alpha$-corruption model is $O(\frac{\epsilon}{(\epsilon - \alpha)^2})$-competitive in expectation.*

Similar to the random sample model, we have an almost matching lower bound for the $\epsilon$-random sample with $\alpha$-corruption model under the mild assumption that the corruption is at most half of the whole sample. The proof of the following theorem is similar to Theorem 2 and can be found in Section B.

**Theorem 5.** *For any $\epsilon \in (0,1)$ and $\alpha \in (0, \frac{\epsilon}{2})$, every algorithm in the $\epsilon$-random sample with $\alpha$-corruption model has competitive ratio $\Omega(\frac{1}{\epsilon - \alpha})$.*

**Proof Overview of Theorem 4:** The proof of Theorem 4 has the same structure as Theorem 1; in particular, we split the cost of semi-online Pivot into an offline- and online phase. The offline- and online phases roughly correspond to $S$ and $V \setminus S$, respectively. However, the corruptions introduce new challenges in both cases.

For the offline phase, we can no longer argue that $S$ is distributed as the first $\epsilon n$ arrivals in the random order model due to the adversarially chosen part of $S$. Instead, we prove a generalization of Theorem 3 via dual fitting that allows us to handle more general arrival distributions. Roughly, to bound the cost of the offline phase, it suffices to show that if $A_t$ occurs in the offline phase, then each vertex of $t$ has reasonable probability to be the pivot that "causes" $A_t$.

For the online phase, we still argue by Lemma 2 that it suffices to show that the remaining vertices in the online phase have small positive degree. Here, we prove that even with adversarial corruptions, running Pivot in the offline phase still sparsifies the remaining graph.

## 5    Experiments

In this section, we empirically validate our theoretical findings. Specifically, we:

- Demonstrate that Pivot's performance can be poor when the arrival sequence is chosen adversarially.
- Verify that with a small portion of nodes randomly drawn from the dataset (which we refer to as **advice**), semi-online Pivot is competitive with offline Pivot in random order (which is a 3-approximation to optimum). Further, we show that the result is robust across multiple parameter settings and additional corruptions.
- Show that when the advice is temporal (and not random) the performance is still strong.

**Datasets.**    We use the following datasets from the Stanford Large Network Dataset Collection [Leskovec and Krevl, 2014][1]. Each dataset is a social network where the nodes represent entities

---

[1] https://snap.stanford.edu/data/, *ego-Facebook*: https://snap.stanford.edu/data/ego-Facebook.html, *ego-Gplus*: https://snap.stanford.edu/data/ego-Gplus.html, *soc-RedditHyperlinks*: https://snap.stanford.edu/data/soc-RedditHyperlinks.html, *soc-sign-bitcoin-otc*: https://snap.stanford.edu/data/soc-sign-bitcoin-otc.html

such as users or communities, and the edges represent connections between the entities such as communication or transactions. We use two non-temporal datasets: *ego-Facebook* and *ego-Gplus* [McAuley and Leskovec, 2012]; and two temporal datasets: *soc-RedditHyperlinks* [Kumar et al., 2018a] and *soc-sign-bitcoin-otc* [Kumar et al., 2016, 2018b]. The temporal datasets have *timestamps* on the edges representing when the connection happened. We refer to these datasets as FACEBOOK, GPLUS, REDDIT and BITCOIN respectively. See Table 1 for a full description of the four data sources.

To generate signed complete graphs, on directed graphs we convert all arcs into undirected edges, and on weighted graphs we only keep the edges with positive weights. These edges represent the positive edges, and all the other edges are negative edges.

Table 1: Dataset desciption.

| Dataset | (# Edges, # Nodes) | Edge Features | Edge Meaning |
|---|---|---|---|
| FACEBOOK | (4039, 88234) | Undirected, unweighted, non-temporal | Users are friends |
| GPLUS | (107614, 13673453) | Directed, unweighted, non-temporal | One user follows another |
| REDDIT | (55863, 858490) | Directed, weighted, temporal | A hyperlink between two subreddits |
| BITCOIN | (5881, 35592) | Directed, weighted, temporal | A trust rating between users |

To perform a richer evaluation, we sub-sample from the datasets to obtain our input graphs in the following manner. The two non-temporal data sources, FACEBOOK and GPLUS, contain *ego-networks* which represent the lists of friends of pre-selected users and the connections among them. We experiment on these ego-networks which are sub-graphs induced from the entire graph by a subset of nodes. For the temporal data sources, we take all edges with timestamps falling into a time interval and all nodes induced on these edges.

**Performance Metrics.** We use Pivot in uniformly random order as our benchmark. For each arrival order constructed, we conduct multiple trials, take the ratio of the average number of *disagreements* for semi-online Pivot over that of random Pivot, and subtract 1 to measure relative improvement, which we call **degradation**.

**Power of Advice.** Our first experiment studies the benefit of a random sample of nodes. We compare the performance of Pivot with and without advice under different element orderings.

We first test Pivot on a uniform random ordering of the nodes. Recall that this is a 3-approximation [Ailon et al., 2008] to the optimum. Then we test Pivot in the following adversarial orders: (1) node degree in descending order; (2) number of bad triangles in a node's neighborhood that contains this node, in descending order[2]; and (3) timestamp, in chronological order (only for temporal datasets.) [3]

Next, we explore the performance of *semi-online* Pivot with advice in the $\epsilon$-random sample model. We produce the advice by randomly sampling an $\epsilon$-fraction of the nodes. Then, in the online phase, the remaining nodes arrive in adversarial order. We test the same adversarial orders as before.

Table 2: Mean degradation and standard deviation in Pivot's performance on different sequences, $\epsilon = 0.1$, 30 trials.

| Dataset | (# Nodes, # Edges) | Random | Degree | Degree w. Advice | Bad Triangles | Bad Triangles w. Advice | Time | Time w. Advice |
|---|---|---|---|---|---|---|---|---|
| FACEBOOK | (534, 9626) | 0 ± 11.21 % | 108.61 % | 4.33 ± 12.68 % | 108.54 % | 3.53 ± 13.46 % | N/A | N/A |
| FACEBOOK | (1034, 53498) | 0 ± 6.00 % | 63.19 % | 0.19 ± 5.49 % | 63.72 % | 1.48 ± 6.63 % | N/A | N/A |
| GPLUS | (1650, 166292) | 0 ± 20.79 % | 133.46 % | 9.90 ± 23.30 % | 133.46 % | -0.80 ± 17.03 % | N/A | N/A |
| GPLUS | (3455, 435569) | 0 ± 15.35 % | 229.37 % | -3.95 ± 16.03 % | 229.38 % | -3.91 ± 12.78 % | N/A | N/A |
| REDDIT | (4277, 9524) | 0 ± 42.79 % | 610.64 % | 3.99 ± 8.31 % | 610.97 % | 5.71 ± 8.49 % | 139.26 % | -4.88 ± 10.43% |
| REDDIT | (7019, 20724) | 0 ± 11.36 % | 690.05 % | 21.28 ± 28.61 % | 690.27 % | 18.40 ± 11.74 % | 79.89% | 4.59 ± 8.65 % |
| REDDIT | (14042, 56567) | 0 ± 20.23 % | 822.60 % | 11.09 ± 42.99 % | 822.64 % | 2.59 ± 12.24 % | 50.24 % | -1.01 ± 14.11 % |
| BITCOIN | (2979, 14695) | 0 ± 21.83 % | 1070.07 % | 9.75 ± 24.32 % | 1066.81 % | 11.70 ± 25.48 % | 579.06 % | 8.50 ± 40.54 % |

See Table 2 for data on the performance of Pivot in different scenarios. Each entry shows the level of degradation compared with fully random Pivot after 30 trials (mean ± standard deviation). See Section C for results on all input graphs. The first two columns show the data source and the size of the graph instance. Column "Random" shows random Pivot's performance. Columns "Degree", "Bad Triangles" and "Time" give Pivot's performance on the three adversarial orders mentioned earlier, while the columns next to these three show the performance of semi-online Pivot using the advice.

---

[2]For a node with $k$ neighbors, this number is equivalent to its **clustering coefficient** times $\frac{k(k-1)}{2}$.

[3]The original temporal datasets only have timestamps on edges. We consider the timestamp of any node to be the earliest timestamp of all edges adjacent to it.

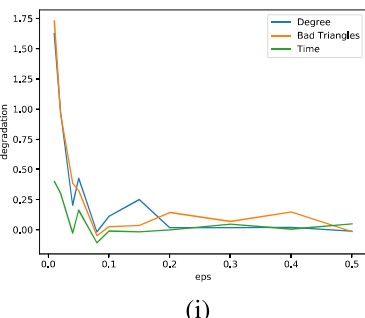
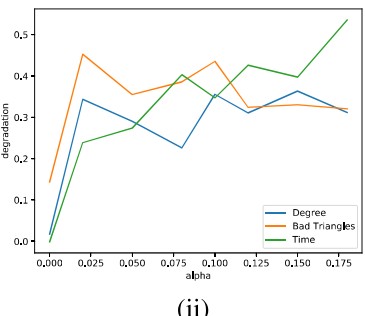

(i)                                   (ii)

Figure 1:   (i) Degradation v.s. $\epsilon$ values, REDDIT, (# nodes, # edges) = (14042, 56567). (ii) Degradation v.s. $\alpha$ values, when $\epsilon = 0.2$.

Compared with a fully random sequence, an adversarial sequence of online node arrivals can cause Pivot's performance to degrade significantly, from 2-10x. Sorting the nodes according to degrees or bad triangles in the neighborhood could causes Pivot's performance to be $1000\%$ worse for some sub-samples. For the temporal node sequences, Pivot's performance is about $50 - 100\%$ worse for REDDIT and $500\%$ worse for BITCOIN. However, when $\epsilon = 0.1$, semi-online Pivot's performance is only slightly worse (about $10 - 20\%$) than fully random Pivot. Thus, with only a small advice set, semi-online Pivot almost completely bridges the gap between adversarial and random order.

**Robustness.** Our next goal is to show that the gains of Pivot with advice are robust to different choices of $\epsilon$ as well as adversarial corruptions. Recall that for a node set of size $n$, the advice has $(\epsilon - \alpha)n$ randomly sampled nodes and $\alpha n$ adversarially chosen nodes. For each of the four data sources, we select a sufficiently large sub-sample and change $\epsilon$ and $\alpha$ while producing advice.

For the corrupted advice, for any one of the three adversarial sequences, we choose the first $\alpha$ fraction of nodes to be the corruptions (i.e. the "worst" nodes as evidenced by the experiments with no advice). Then, we sample $(\epsilon - \alpha)n$ nodes from the remaining nodes to complete the advice. The online phase arrival order is the same as in the adversarial sequence.

Figure 1(i) shows how degradation changes while $\epsilon$ ranges in $[0.01, 0.5]$ and $\alpha = 0$ (no corruptions), and (ii) shows how corruptions in the advice affect semi-online Pivot when $\epsilon = 0.2$. Pivot's performance is robust against shrinking the size of the advice and against adding corruptions. For $\epsilon \geq 0.05$, or with the advice set corrupted by $25\%$, the performance remains constant. See Section C for results on more datasets.

**Temporal Advice.** Finally, learning the advice from prior temporal data is sufficient for Pivot to have good performance. We test both temporal datasets: REDDIT and BITCOIN. The advice is generated from older data just prior to the time interval of the test dataset. In particular, our advice consists of all nodes in the older dataset, which are incident on edges in the test dataset (i.e. the old data that is related to the new data.) After the advice, the test data set arrives in adversarial order - including temporal order, where vertices arrive in order of their timestamps.

Table 3: Degradation of semi-online Pivot when using advice from historical data, REDDIT, 100 trials, test dataset has the duration of 6 months.

| Days | $\epsilon$-value | Random | Degree | Degree w. Advice | Bad Triangles | Bad Triangles w. Advice | Time | Time w. Advice |
|---|---|---|---|---|---|---|---|---|
| 5 | 0.09 | $0 \pm 14.57\%$ | $675.40\%$ | $60.73 \pm 38.29\%$ | $675.12\%$ | $65.73 \pm 46.47\%$ | $52.08\%$ | $55.49 \pm 29.57\%$ |
| 10 | 0.15 | $0 \pm 14.57\%$ | $675.40\%$ | $42.09 \pm 25.19\%$ | $675.12\%$ | $36.85 \pm 20.26\%$ | $52.08\%$ | $34.74 \pm 23.39\%$ |
| 15 | 0.19 | $0 \pm 14.57\%$ | $675.40\%$ | $41.09 \pm 32.35\%$ | $675.12\%$ | $36.85 \pm 20.26\%$ | $52.08\%$ | $36.58 \pm 37.41\%$ |
| 20 | 0.22 | $0 \pm 14.57\%$ | $675.40\%$ | $30.76 \pm 23.27\%$ | $675.12\%$ | $36.43 \pm 32.97\%$ | $52.08\%$ | $41.66 \pm 48.21\%$ |
| 25 | 0.25 | $0 \pm 14.57\%$ | $675.40\%$ | $35.28 \pm 50.72\%$ | $675.12\%$ | $30.97 \pm 32.24\%$ | $52.08\%$ | $30.54 \pm 28.82\%$ |

Table 3 shows the results for REDDIT when we change the time interval used to generate the advice, which is shown in Column "Days" in time unit of days. The time interval for the *test dataset* is fixed to be 6 months. Column "$\epsilon$-value" gives the proportion of nodes in test dataset that appeared in the old data, which corresponds to the parameter $\epsilon$ in the $\epsilon$-random sample model. See Section C for the results for BITCOIN. The rest of the table shows the degradation of Pivot using the advice from the corresponding historical data. Pivot's performance improves when we increase the time interval (equivalent to increasing $\epsilon$). Starting with $\epsilon = 0.15$ (time interval for old data/test data is 10

days/6 months), the temporal advice improves Pivot's performance on all adversarial orders. This demonstrates that Pivot works well empirically using advice from historical data.

## 6   Conclusion

By augmenting a standard online algorithm for correlation clustering with a small random sample of the data, we can overcome strong lower bounds for online correlation clustering. We give near-optimal algorithms for semi-online correlation clustering given a random sample (with corruptions), and further this theory is predictive of practical performance. Empirically, with only a small random sample, semi-online pivot is competitive with *offline* pivot in random order. Further, in temporal datasets, the sample can be practically obtained from past data. We show that semi-online models, where we augment an online algorithm with some offline information, can be a powerful tool in both theory and practice to improve the performance of online algorithms, and we believe they will find further applications in other problems.

## Funding Sources

B. Moseley, Y. Wang and R. Zhou were supported in part by a Google Research Award, an Infor Research Award, a Carnegie Bosch Junior Faculty Chair and NSF grants CCF-1824303, CCF-1845146, CCF-1733873 and CMMI-1938909. B. Moseley additionally is a part time employee of Relational AI.

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
