# A Analysis: Proof of Theorem 4

Throughout this section, let OPT denote the number of disagreements made by the optimal offline algorithm on $G$ and ALG denote the cost of semi-online Pivot. Note that OPT is a fixed quantity, while ALG depends on the random choice of $S$ and its order. Further, all probabilities are taken with respect to the $\epsilon$-random sample with $\alpha$-corruption model.

By Equation (1), we can write our cost as:

$$\mathbb{E}[\text{ALG}] = \sum_{t \in T} \mathbb{P}(A_t).$$

Previously, we split this sum into the contributions by $S$ and $V \setminus S$. Now, we consider a slightly different partition. To this end, we define the (random) ordered set $S' \subset S$ of the first $\frac{\epsilon - \alpha}{2}n$ arrivals in $S$. Recall that $A_t^{S'} = \{$one vertex of $t$ is chosen as pivot in $S'$ while all three are unclustered$\}$ and analogously for $A_t^{V \setminus S}$. Then we can re-write:

$$\mathbb{E}[Alg] = cost(S') + cost(V \setminus S'),$$

where $cost(S') = \sum_{t \in T} \mathbb{P}(A_t^{S'})$ and $cost(V \setminus S') = \sum_{t \in T} \mathbb{P}(A_t^{V \setminus S'})$. Thus, now our offline phase corresponds to $S'$, and our online phase to $V \setminus S'$. We will show that $cost(S') = O(\frac{1}{\epsilon - \alpha})$OPT and $cost(V \setminus S') = O(\frac{\epsilon}{(\epsilon - \alpha)^2})$OPT. Combining these two bounds with the above expression for $\mathbb{E}[\text{ALG}]$ completes the proof of Theorem 4.

### A.0.1 Bounding Cost of $S'$

We show that $cost(S') = O(\frac{1}{\epsilon - \alpha})$OPT. Recall that $S'$ is part of the offline phase which arrives in random order. For any vertex $i \in V$, we define the event $A_i^{S'}$ analogously as $A_t^{S'}$ for triangle $t$. That is, $A_i^{S'} = \{i$ is chosen as pivot in $S'\}$. We begin with a technical lemma that analogous to the dual fitting analysis of Pivot in random order.

**Lemma 4.** *Let $c > 0$. Suppose for all $t \in T$ and $i \in t$, we have $\mathbb{P}(A_i^{S'} \mid A_t^{S'}) \geq c$. Then $\sum_{t \in T} \mathbb{P}(A_t^{S'}) \leq \frac{1}{c}$OPT.*

*Proof.* Consider the primal-dual pair of linear programs $P$ and $D$:

$$P = \min_x \{\sum_{e \in E} x_e \mid x_{ij} + x_{jk} + x_{ki} \geq 1 \, \forall ijk \in T, \, x \geq 0\}.$$

$$D = \max_y \{\sum_{t \in T} y_t \mid \sum_{t \in T \mid e \in t} y_t \leq 1 \, \forall e \in E, \, y \geq 0\}.$$

We observe that $x_e = 1_{Opt \text{ makes a disagreement on } e}$ for all $e$ is feasible for $P$ with objective value OPT. Thus, OPT$(P) \leq$ OPT. Further, by strong duality OPT$(D) =$ OPT$(P) \leq$ OPT.

We now exhibit a feasible solution to $D$. For all $t \in T$, take $y_t = c\mathbb{P}(A_t^{S'})$. It suffices to show that this setting of $y$ is feasible for $D$, because then $c \sum_{t \in T} \mathbb{P}(A_t^{S'}) \leq$ OPT. It is immediate that $y \geq 0$, so we check for each $ij \in E$:

$$\sum_{k \in V \mid ijk \in T} c\mathbb{P}(A_{ijk}^{S'}) \leq \sum_{k \in V \mid ijk \in T} \mathbb{P}(A_k^{S'} \mid A_{ijk}^{S'})\mathbb{P}(A_{ijk}^{S'}) = \sum_{k \mid ijk \in T} \mathbb{P}(A_k^{S'} \cap A_{ijk}^{S'}),$$

where we observe that the events $A_k^{S'} \cap A_{ijk}^{S'}$ are disjoint for all $k$ with $ijk \in T$, and each such event implies that Pivot makes $ij$ a disagreement. We conclude:

$$\sum_{k \in V \mid ijk \in T} c\mathbb{P}(A_{ijk}^{S'}) \leq \mathbb{P}(\text{Pivot makes } ij \text{ a disagreement}) \leq 1.$$

Thus, $y$ is feasible for $D$. $\qquad\qquad\square$

Note that we can recover the proof of Theorem 3 using the above lemma. Take all probabilities with respect to the random order model, and $S' = V$. Then $\mathbb{P}(A_i^V \mid A_t^V) \geq \frac{1}{3}$ for all $t \in T$ and $i \in t$. This is because each vertex of $t$ has equal probability of arriving first among the three in the random order model. This implies the expected cost of Pivot in the random order model is at most $3\text{OPT}$.

In light of the above lemma, we study the probabilities $\mathbb{P}(A_i^{S'} \mid A_t^{S'})$ for $t \in T, i \in t$. Roughly, we want to show that the probability of each vertex of $t$ arriving first among the three is not too small. The case to keep in mind is if some vertices of a bad triangle are adversarial and others are not. The next lemma shows that the probability than the next arrival in $S'$ is a particular vertex does not vary much between adversarial and random vertices.

**Lemma 5.** *Let $r \leq |S'|$. Fix an ordered prefix $p = (p_1, \ldots, p_{r-1})$ of the first $r - 1$ arrivals in $S'$. Then for any $i \notin p$, we have:*

$$\frac{\epsilon - \alpha}{2} \frac{1}{\epsilon n - (r-1)} \leq \mathbb{P}(r\text{th arrival is } i \mid \text{first } r - 1 \text{ arrivals are } p) \leq \frac{1}{\epsilon n - (r-1)}.$$

*Proof.* Fix $i \notin p$ and let $P$ be the event $P = \{\text{first } r - 1 \text{ arrivals are } p\}$. Noting that $i \in S$ is necessary for $i$ to be the $r$th arrival, we have:

$$\mathbb{P}(r\text{th arrival is } i \mid P) = \mathbb{P}(r\text{th arrival is } i \mid P, i \in S)\mathbb{P}(i \in S \mid P).$$

Conditioned on $P$ and $i \in S$, the $r$th arrival is a uniform random vertex from $S \setminus P$, so:

$$\mathbb{P}(r\text{th arrival is } i \mid P, i \in S) = \frac{1}{|S| - |P|} = \frac{1}{\epsilon n - (r-1)}.$$

To analyze the second term, $\mathbb{P}(i \in S \mid P)$, we consider two cases. In the first case $i \in A$, so $\mathbb{P}(i \in S \mid P) = 1$. Otherwise, $i \in V \setminus A$, so conditioned on $P$, $i \in S$ if and only if $i \in R \setminus \{p_1, \ldots, p_{r-1}\}$. Thus we can lower-bound $\mathbb{P}(i \in S \mid P) \geq \frac{(\epsilon-\alpha)n - r}{n} \geq \frac{\epsilon - \alpha}{2}$, where we recall $r \leq |S'| = \frac{(\epsilon-\alpha)n}{2}$. We conclude, $\frac{\epsilon-\alpha}{2} \leq \mathbb{P}(i \in S \mid P) \leq 1$. Combining our expressions for both terms gives the desired result. $\square$

Now we are ready to lower bound $\mathbb{P}(A_i^{S'} \mid A_t^{S'})$ by considering the prefixes of arrivals where $A_t^{S'}$ occurs.

**Lemma 6.** *For any $t \in T$ and $i \in t$, $\mathbb{P}(A_i^{S'} \mid A_t^{S'}) \geq \frac{\epsilon-\alpha}{5}$.*

*Proof.* Fix $t \in T$ and $i \in t$. For $1 \leq r \leq |S'|$, we define the event $A_t^r = \{\text{one vertex of } t \text{ is chosen as pivot in the } r\text{th arrival while all three are unclustered}\}$. Note that the events $A_t^r$ for all $1 \leq r \leq |S'|$ partition $A_t^{S'}$. Then by the law of total probability:

$$\mathbb{P}(A_i^{S'} \mid A_t^{S'}) = \sum_{r=1}^{|S'|} \mathbb{P}(A_i^{S'} \mid A_t^{S'}, A_t^r)\mathbb{P}(A_t^r \mid A_t^{S'}) = \sum_{r=1}^{|S'|} \mathbb{P}(r\text{th arrival is } i \mid A_t^r)\mathbb{P}(A_t^r \mid A_t^{S'}).$$

Because $\sum_{r=1}^{|S'|} \mathbb{P}(A_t^r \mid A_t^{S'}) = 1$, it suffices to lower bound $\mathbb{P}(r\text{th arrival is } i \mid A_t^r) \geq \frac{\epsilon-\alpha}{5}$ for all $1 \leq r \leq |S'|$.

To this end, we fix any $1 \leq r \leq |S'|$, and let $t = ijk$. Noting that the events that the $r$th arrival is $i$, $j$, or $k$ partition $A_t^r$, we have:

$$\mathbb{P}(r\text{th arrival is } i \mid A_t^r) + \mathbb{P}(r\text{th arrival is } j \mid A_t^r) + \mathbb{P}(r\text{th arrival is } k \mid A_t^r) = 1$$

$$\mathbb{P}(r\text{th arrival is } i \mid A_t^r) = \left(1 + \frac{\mathbb{P}(r\text{th arrival is } j \mid A_t^r)}{\mathbb{P}(r\text{th arrival is } i \mid A_t^r)} + \frac{\mathbb{P}(r\text{th arrival is } k \mid A_t^r)}{\mathbb{P}(r\text{th arrival is } i \mid A_t^r)}\right)^{-1}.$$

It remains to upper bound the ratio $\frac{\mathbb{P}(r\text{th arrival is } u \mid A_t^r)}{\mathbb{P}(r\text{th arrival is } v \mid A_t^r)}$ for $u, v \in t$. To this end, we define the set $P$ of all ordered prefixes $p$ of $r - 1$ arrivals such that after running Pivot on $p$, all vertices in $t$ are uncovered. In particular, $A_t^r$ occurs if any only if the first $r - 1$ arrivals is some $p \in P$ and the $r$th

arrival is in $t$. Then for any $u \in t$:

$$\mathbb{P}(r\text{th arrival is } u \mid A_t^r) = \frac{\mathbb{P}(r\text{th arrival is } u, A_t^r)}{\mathbb{P}(A_t^r)}$$

$$= \frac{\sum_{p \in P} \mathbb{P}(r\text{th arrival is } u, \text{first } r-1 \text{ arrivals are } p)}{\mathbb{P}(A_t^r)}$$

$$= \frac{\sum_{p \in P} \mathbb{P}(r\text{th arrival is } u \mid \text{first } r-1 \text{ arrivals are } p)\mathbb{P}(\text{first } r-1 \text{ arrivals are } p)}{\mathbb{P}(A_t^r)}.$$

Now consider $u, v \in t$. Applying the upper- and lower bounds given by Lemma 5, respectively, we have:

$$\mathbb{P}(r\text{th arrival is } u \mid A_t^r) \leq \frac{1}{\epsilon n - (r-1)} \sum_{p \in P} \frac{\mathbb{P}(\text{first } r-1 \text{ arrivals are } p)}{\mathbb{P}(A_t^r)},$$

$$\mathbb{P}(r\text{th arrival is } v \mid A_t^r) \geq \frac{\epsilon - \alpha}{2} \frac{1}{\epsilon n - (r-1)} \sum_{p \in P} \frac{\mathbb{P}(\text{first } r-1 \text{ arrivals are } p)}{\mathbb{P}(A_t^r)},$$

which taken together imply $\frac{\mathbb{P}(r\text{th arrival is } u \mid A_t^r)}{\mathbb{P}(r\text{th arrival is } v \mid A_t^r)} \leq \frac{2}{\epsilon - \alpha}$. We conclude:

$$\mathbb{P}(r\text{th arrival is } i \mid A_t^r) \geq \left(1 + \frac{2}{\epsilon - \alpha} + \frac{2}{\epsilon - \alpha}\right)^{-1} = \frac{\epsilon - \alpha}{4 + \epsilon - \alpha} \geq \frac{\epsilon - \alpha}{5}.$$

$\square$

Composing Lemma 4 and Lemma 6 gives $cost(S') = \sum_{t \in T} \mathbb{P}(A_t^{S'}) \leq \frac{5}{\epsilon - \alpha}\textsc{Opt} = O(\frac{1}{\epsilon - \alpha})\textsc{Opt}$.

### A.0.2   Bounding Cost of $V \setminus S'$

Here we show that $cost(V \setminus S') = O(\frac{\epsilon}{(\epsilon - \alpha)^2})\textsc{Opt}$. Our strategy again is to show that Pivot on $S'$ sparsifies the remaining graph, so every arrival in $V \setminus S'$ has a small number of positive unclustered neighbors. Then we apply Lemma 2. The next lemma is analogous to Lemma 3, but the estimates are slightly more involved due to the more complex arrival distribution.

**Lemma 7.** *For any $v \in V$, define the random variable $N_v$ to be $0$ if $v$ is clustered by running Pivot on $S'$ (i.e. $v \in S'$ or $v$ has a positive edge to a pivot in $S'$) or $N_v = |\{unclustered\ positive\ neighbors\ of\ v\ in\ V \setminus S'\}|$ otherwise. Then $\mathbb{E}N_v = O(\frac{\epsilon}{(\epsilon - \alpha)^2})$.*

*Proof.* Fix $v \in V$ and $k \geq 0$. We first upper bound $\mathbb{P}(N_v \geq k)$. Define the event $C_i = \{$v is unclustered and has at least $k$ positive unclustered neighbors after $i$th arrival in $S'\}$. Then $\mathbb{P}(N_v \geq k) = \mathbb{P}(C_{|S'|} \mid C_{|S'|-1}, \ldots, C_1) \ldots \mathbb{P}(C_1)$. For all $i$, we have:

$$\mathbb{P}(C_i \mid C_{i-1}, \ldots, C_1) \leq 1 - \mathbb{P}(v \text{ clustered by } i\text{th arrival} \mid C_{i-1}, \ldots, C_1).$$

It is convenient to describe the distribution of the $i$th arrival in $S'$ as follows: Suppose the first $i-1$ arrivals consist of random arrivals $R' \subset V \setminus A$ and adversarial arrivals $A' \subset A$. First, we decide whether the $i$th arrival will be random (from $V \setminus A$) or adversarial (from $A$.) It is random with probability $\frac{(\epsilon - \alpha)n - |R'|}{\epsilon n - i + 1} \geq \frac{(\epsilon - \alpha)n - |R'|}{\epsilon n} \geq \frac{\epsilon - \alpha}{2\epsilon}$, using the fact that $|R'| \leq |S'| = (\epsilon - \alpha)n/2$. Similarly, the $i$th arrival is adversarial with probability $\frac{\alpha n - |A'|}{\epsilon n - i + 1} \geq \frac{\alpha n - |A'|}{\epsilon n}$. After deciding whether the $i$th arrival will be random or adversarial, then we choose the arrival by drawing a uniform random vertex from $(V \setminus A) \setminus R'$ or $A \setminus A'$, respectively.

Now we lower bound $\mathbb{P}(v \text{ clustered by } i\text{th arrival} \mid C_{i-1}, \ldots, C_1)$. Conditioned on $C_{i-1}$, $v$ has at least $k$ unclustered positive neighbors before the $i$th arrival. Let $r$ and $a$ denote the number of unclustered positive neighbors in $V \setminus A$ and $A$, respectively. Note that $r + a \geq k$. Then we compute:

$$\mathbb{P}(v \text{ clustered by } i\text{th arrival} \mid C_{i-1}, \ldots, C_1) \geq \frac{\epsilon - \alpha}{2\epsilon}\frac{r}{n} + \frac{\alpha n - |A'|}{\epsilon n}\frac{a}{\alpha n - |A'|}$$

$$= \frac{\epsilon - \alpha}{2\epsilon}\frac{r}{n} + \frac{1}{\epsilon}\frac{a}{n}$$

$$\geq \frac{\epsilon - \alpha}{2\epsilon}\frac{k}{n}.$$

Recalling $|S'| = (\epsilon - \alpha)n/2$, we have:

$$\mathbb{P}(N_v \geq k) \leq \left(1 - \frac{\epsilon - \alpha}{2\epsilon}\frac{k}{n}\right)^{(\epsilon - \alpha)n/2} \leq \exp\left(-\frac{(\epsilon - \alpha)^2 k}{4\epsilon}\right).$$

Using this tail bound, we can bound the expectation of $N_v$:

$$\mathbb{E}N_v = \sum_{k=0}^{\infty}\mathbb{P}(N_v \geq k) \leq \sum_{k=0}^{\infty}\exp\left(-\frac{(\epsilon - \alpha)^2 k}{4\epsilon}\right) = O\left(\int_0^{\infty}\exp\left(-\frac{(\epsilon - \alpha)^2 x}{4\epsilon}\right)dx\right) = O\left(\frac{\epsilon}{(\epsilon - \alpha)^2}\right).$$

$\square$

Composing Lemma 2 and Lemma 7, we can bound $cost(V \setminus S')$. Let $G'$ denote the subgraph of $G$ induced by all vertices in $V \setminus S'$ that are not clustered by the pivots chosen in $S'$. Then we have the same three properties as in the proof of Theorem 4, which we repeat for convenience:

- Let $T'$ be the set of all bad triangles in $G'$. We have $\sum_{t \in T} 1_{A_t^{V \setminus S'}} \leq |T'|$.
- The positive degree in $G'$ of a vertex $v$ is exactly $N_v$ (as defined in Lemma 7.)
- Let $E'$ denote the edge set of $G'$ and $E^*$ the disagreements made by OPT. Then OPT induces a clustering of $G'$ that makes disagreements $E' \cap E^*$.

Using Lemma 2 and Lemma 7 and the above three properties, we conclude:

$$cost(V \setminus S') \leq \mathbb{E}|T'| \leq \mathbb{E}\left[\sum_{ij \in E' \cap E^*} d_{G'}^+(i) + d_{G'}^+(j)\right] \leq \sum_{ij \in E^*}\mathbb{E}N_i + \mathbb{E}N_j = O\left(\frac{\epsilon}{(\epsilon - \alpha)^2}\right)\text{OPT}.$$

## B  Omitted Proofs

*Proof of Lemma 3.* Fix $v \in V$ and $k \geq 0$. We first upper bound $\mathbb{P}(N_v \geq k)$. Define the event $C_i = \{$v is unclustered and has at least $k$ positive unclustered neighbors after $i$th arrival in $S\}$. Then $\mathbb{P}(N_v \geq k) = \mathbb{P}(C_{|S|} \mid C_{|S|-1}, \ldots, C_1)\ldots\mathbb{P}(C_1)$. For all $i$, we have:

$$\mathbb{P}(C_i \mid C_{i-1}, \ldots, C_1) \leq 1 - \mathbb{P}(\text{v clustered by } i\text{th arrival} \mid C_{i-1}, \ldots, C_1) \leq 1 - \frac{k+1}{n-(i-1)} \leq 1 - \frac{k}{n},$$

where in the second inequality, we use the fact that conditioned on $C_{i-1}$, $v$ has at least $k$ unclustered neighbors before the $i$th arrival. Recalling $|S| = \epsilon n$, we have $\mathbb{P}(N_v \geq k) \leq (1 - \frac{k}{n})^{\epsilon n} \leq \exp(-\epsilon k)$. Using this tail bound, we can bound the expectation of $N_v$: $\mathbb{E}N_v = \sum_{k=0}^{\infty}\mathbb{P}(N_v \geq k) \leq \sum_{k=0}^{\infty}\exp(-\epsilon k) = O(\int_0^{\infty}\exp(-\epsilon k)dk) = O(\frac{1}{\epsilon})$. $\square$

*Proof of Theorem 5.* It suffices to consider $\epsilon - \alpha < \frac{1}{2}$. Then $\epsilon \in (0, \frac{3}{4})$. Consider any sufficiently large $n$ such that $\frac{1}{\epsilon - \alpha} + \epsilon n \leq n$. Then we define the graph on $n$ vertices. All edges are negative except those on a set $L$ of $\frac{1}{\epsilon - \alpha}$ vertices. The edges on $L$ are a lower bound instance of size $\frac{1}{\epsilon - \alpha}$ guaranteed by Lemma 1.

The adversary chooses the corrupted nodes of the sample to be those not in $L$. Let $R$ be the corrupted random sample given to the algorithm. Then the probability that $R$ contains no vertex from $L$ is:

$$\Pr(R \cap L = \emptyset) = (1 - \frac{1/(\epsilon - \alpha)}{(1-\alpha)n})\ldots(1 - \frac{1/(\epsilon - \alpha)}{(1-\alpha)n - (\epsilon - \alpha)n + 1}) \geq (1 - \frac{1/(\epsilon - \alpha)}{n - \epsilon n})^{(\epsilon - \alpha)n} = \Omega(1).$$

Further, conditioned on this event, the cost of any algorithm is $\Omega(\frac{1}{\epsilon - \alpha})$-competitive, because $L$ arrives in adversarial order. $\square$

## C  Experiments

This section contains the numerical results omitted in Section 5.

Table 4: Degradation in Pivot's performance on different sequences, $\epsilon = 0.1$, 30 trials, data source FACEBOOK

| (# Nodes, # Edges) | Random | Degree | Degree w. Advice | Bad Triangles | Bad Triangles w. Advice |
|---|---|---|---|---|---|
| (168, 3312) | $0 \pm 15.09\,\%$ | $57.72\,\%$ | $-0.16 \pm 11.28\,\%$ | $57.72\,\%$ | $1.67 \pm 14.45\,\%$ |
| (333, 5038) | $0 \pm 12.10\,\%$ | $40.73\,\%$ | $1.90 \pm 11.23\,\%$ | $41.59\,\%$ | $1.59 \pm 10.16\,\%$ |
| (534, 9626) | $0 \pm 11.21\,\%$ | $108.61\,\%$ | $4.33 \pm 12.68\,\%$ | $108.54\,\%$ | $3.53 \pm 13.46\,\%$ |
| (747, 60050) | $0 \pm 13.28\,\%$ | $99.45\,\%$ | $0.95 \pm 11.09\,\%$ | $101.65\,\%$ | $-0.29 \pm 11.39\,\%$ |
| (1034, 53498) | $0 \pm 6.00\,\%$ | $63.19\,\%$ | $0.19 \pm 5.49\,\%$ | $63.72\,\%$ | $1.48 \pm 6.63\,\%$ |

Table 5: Degradation in Pivot's performance on different sequences, $\epsilon = 0.1$, 30 trials, data source GPLUS

| (# Nodes, # Edges) | Random | Degree | Degree w. Advice | Bad Triangles | Bad Triangles w. Advice |
|---|---|---|---|---|---|
| (638, 16043) | $0 \pm 12.13\,\%$ | $95.67\,\%$ | $-3.40 \pm 12.12\,\%$ | $95.09\,\%$ | $-1.78 \pm 8.43\,\%$ |
| (780, 26552) | $0 \pm 20.27\,\%$ | $160.87\,\%$ | $1.89 \pm 20.37\,\%$ | $160.98\,\%$ | $-7.01 \pm 8.92\,\%$ |
| (1650, 166292) | $0 \pm 20.79\,\%$ | $133.46\,\%$ | $9.90 \pm 23.30\,\%$ | $133.46\,\%$ | $-0.80 \pm 17.03\,\%$ |
| (2213, 93510) | $0 \pm 14.83\,\%$ | $144.08\,\%$ | $0.24 \pm 13.29\,\%$ | $144.13\,\%$ | $-1.39 \pm 9.40\,\%$ |
| (3455, 435569) | $0 \pm 15.35\,\%$ | $229.37\,\%$ | $-3.95 \pm 16.03\,\%$ | $229.38\,\%$ | $-3.91 \pm 12.78\,\%$ |
| (4586, 352373) | $0 \pm 19.89\,\%$ | $275.07\,\%$ | $1.41 \pm 23.09\,\%$ | $275.07\,\%$ | $-7.79 \pm 12.12\,\%$ |

Table 6: Mean degradation and standard deviation in Pivot's performance on different sequences, $\epsilon = 0.1$, 30 trials, data source REDDIT

| (# Nodes, # Edges) | Random | Degree | Degree w. Advice | Bad Triangles | Bad Triangles w. Advice | Time | Time w. Advice |
|---|---|---|---|---|---|---|---|
| (2502, 4101) | $0 \pm 18.38\,\%$ | $1006.24\,\%$ | $11.56 \pm 20.60\,\%$ | $1006.23\,\%$ | $9.11 \pm 12.15\,\%$ | $111.40\,\%$ | $3.66 \pm 15.76\,\%$ |
| (4277, 9524) | $0 \pm 42.79\,\%$ | $610.64\,\%$ | $3.99 \pm 8.31\,\%$ | $610.97\,\%$ | $5.71 \pm 8.49\,\%$ | $139.26\,\%$ | $-4.88 \pm 10.43\%$ |
| (7019, 20724) | $0 \pm 11.36\,\%$ | $690.05\,\%$ | $21.28 \pm 28.61\,\%$ | $690.27\,\%$ | $18.40 \pm 11.74\,\%$ | $79.89\,\%$ | $4.59 \pm 8.65\,\%$ |
| (10772, 39042) | $0 \pm 8.90\,\%$ | $762.12\,\%$ | $22.57 \pm 40.03\,\%$ | $762.29\,\%$ | $15.23 \pm 13.68\,\%$ | $90.54\,\%$ | $6.21 \pm 13.49\,\%$ |
| (14042, 56567) | $0 \pm 20.23\,\%$ | $822.60\,\%$ | $11.09 \pm 42.99\,\%$ | $822.64\,\%$ | $2.59 \pm 12.24\,\%$ | $50.24\,\%$ | $-1.01 \pm 14.11\,\%$ |

Table 7: Mean degradation and standard deviation in Pivot's performance on different sequences, $\epsilon = 0.1$, 30 trials, BITCOIN

| (# Nodes, # Edges) | Random | Degree | Degree w. Advice | Bad Triangles | Bad Triangles w. Advice | Time | Time w. Advice |
|---|---|---|---|---|---|---|---|
| (330, 964) | $0 \pm 21.57\,\%$ | $256.19\,\%$ | $20.29 \pm 17.42\,\%$ | $256.19\,\%$ | $13.29 \pm 20.86\,\%$ | $170.34\,\%$ | $4.01 \pm 22.11\,\%$ |
| (781, 3119) | $0 \pm 65.01\,\%$ | $423.84\,\%$ | $-0.95 \pm 41.17\,\%$ | $426.73\,\%$ | $3.34 \pm 25.55\,\%$ | $309.53\,\%$ | $5.04 \pm 69.28\,\%$ |
| (1262, 5337) | $0 \pm 14.12\,\%$ | $468.27\,\%$ | $18.55 \pm 17.00\,\%$ | $468.13\,\%$ | $10.61 \pm 7.04\,\%$ | $46.25\,\%$ | $12.60 \pm 41.11\,\%$ |
| (1822, 8624) | $0 \pm 20.24\,\%$ | $914.77\,\%$ | $12.09 \pm 39.00\,\%$ | $914.77\,\%$ | $3.65 \pm 14.49\,\%$ | $500.53\,\%$ | $4.40 \pm 22.94\,\%$ |
| (2979, 14695) | $0 \pm 21.83\,\%$ | $1070.07\,\%$ | $9.75 \pm 24.32\,\%$ | $1066.81\,\%$ | $11.70 \pm 25.48\,\%$ | $579.06\,\%$ | $8.50 \pm 40.54\,\%$ |

**Power of Advice:** Here we show the degradation in Pivot's average performance in adversarial order with and without advice. We consider more sub-samples from all four data sources. See Tables 4, 5, 6 and 7.

**Robustness:** Here we show robustness results for the other three data sources: FACEBOOK, GPLUS and BITCOIN. Notice that adversarial order in time does not apply to the non-temporal data sources FACEBOOK and GPLUS, so the plots for these two datasets have two lines corresponding to ordering according to degree and bad triangles.

In general, the conclusions are similar to those in Section 5. For robustness in the $\epsilon$-random sample model, once $\epsilon$ is sufficiently large (this value is usually about 0.1 for the datasets), the performance of Pivot with advice stays relatively constant, and it is only slightly worse than random Pivot. When there are corruptions, Pivot's performance can go up and down as we fix the value of $\epsilon$ and increase $\alpha$. For example, one might see the degradation increase by $50\%$ as we change $\alpha$ from 0 to $0.5\epsilon$. However, compared with Pivot without advice, in all three adversarial orders we still see significant improvement when we use the corrupted advice regardless, even when $\alpha$ is close to $\epsilon$. See Figures 2, 3, and 4.

**Temporal Advice:** Similar to Table 3 in Section 5, we show the results for BITCOIN. The test dataset has the time interval of 2 years, whereas the historical data is taken from a time interval of length $[20, 40, 60, 80, 100]$ days prior to the time interval of the test dataset. See Table 8.

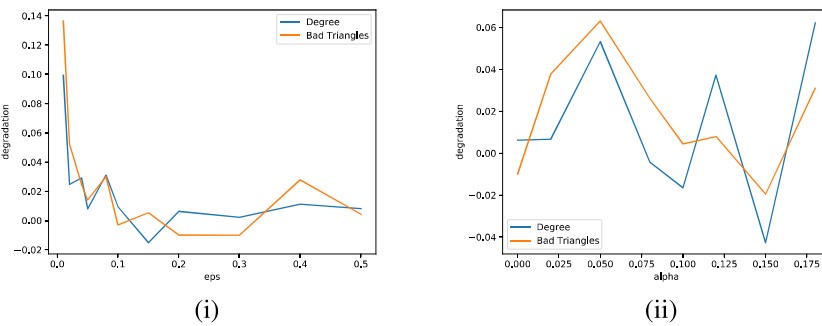

| (i) | (ii) |

Figure 2: (i) Degradation v.s. $\epsilon$ values, FACEBOOK, (# nodes, # edges) = (747, 60050). (ii) Degradation v.s. $\alpha$ values, when $\epsilon = 0.2$.

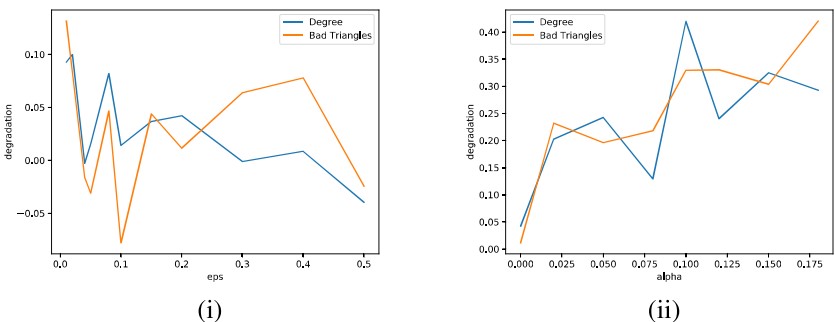

| (i) | (ii) |

Figure 3: (i) Degradation v.s. $\epsilon$ values, GPLUS, (# nodes, # edges) = (4586, 352373). (ii) Degradation v.s. $\alpha$ values, when $\epsilon = 0.2$.

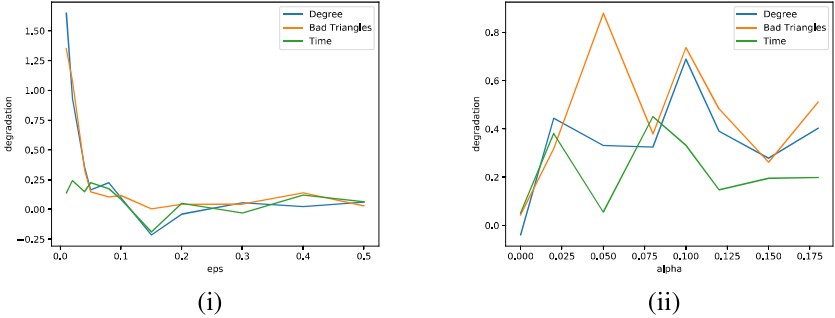

| (i) | (ii) |

Figure 4: (i) Degradation v.s. $\epsilon$ values, BITCOIN, (# nodes, # edges) = (2979, 14695). (ii) Degradation v.s. $\alpha$ values, when $\epsilon = 0.2$.

Table 8: Degradation of semi-online Pivot when using advice from historical data, BITCOIN, 100 trials, test dataset has the duration of 2 years.

| Days | $\epsilon$-value | Random | Degree | Degree w. Advice | Bad Triangles | Bad Triangles w. Advice | Time | Time w. Advice |
|------|------|--------|--------|------------------|---------------|-------------------------|------|----------------|
| 20 | 0.07 | $0 \pm 17.97\%$ | 1069.47 % | $103.11 \pm 155.48\%$ | 1066.21 % | $90.50 \pm 134.53\%$ | 578.71 % | $103.27 \pm 187.77\%$ |
| 40 | 0.11 | $0 \pm 17.97\%$ | 1069.47 % | $65.71 \pm 129.07\%$ | 1066.21 % | $56.87 \pm 113.97\%$ | 578.71 % | $49.55 \pm 125.00\%$ |
| 60 | 0.12 | $0 \pm 17.97\%$ | 1069.47 % | $51.60 \pm 115.25\%$ | 1066.21 % | $67.62 \pm 146.98\%$ | 578.71 % | $28.75 \pm 91.84\%$ |
| 80 | 0.13 | $0 \pm 17.97\%$ | 1069.47 % | $61.90 \pm 140.26\%$ | 1066.21 % | $61.98 \pm 143.11\%$ | 578.71 % | $26.94 \pm 81.91\%$ |
| 100 | 0.14 | $0 \pm 17.97\%$ | 1069.47 % | $59.97 \pm 132.61\%$ | 1066.21 % | $32.51 \pm 79.97\%$ | 578.71 % | $46.28 \pm 124.02\%$ |