# OpenReview forum: "Robust Online Correlation Clustering"
_NeurIPS.cc/2021/Conference — NeurIPS 2021 Poster_

### Official Review · Reviewer_uASx · 2021-06-27

**Rating:** 7
**Confidence:** 3

**Summary:**

The authors of the paper consider an online variant of the well-known Correlation Clustering problem, under the minimum disagreements objective function. Specifically, there is a complete graph $G = (V,E)$ whose edges are partitioned into two sets, namely $E_+$ (positive edges), and $E_-$ (negative edges). In the offline version of the problem, the goal is to find a partition (otherwise called clustering) of $V$ into $C_1, C_2, ..., C_k$, such that the total number of negative edges inside the same cluster plus the total number of positive edges between clusters is minimized. In the online case, the vertices arrive one by one (and upon arrival we are also given the edges between the current vertex and previously arrived ones), the goal is to maintain a clustering at all times, by irrevocably assigning each vertex to a cluster. Here the goal is to achieve the optimal offline value with regards to disagreements.

Although there exist approximation algorithms for the offline version, it is already shown that the online one does not admit any competitive algorithms with meaningful guarantees due to adversarial instances. The main contribution of the paper here is then introducing a semi-online model, in which reasonable competitive algorithms are achievable. This semi-online setting goes as follows.

1) There is an offline phase, in which the algorithm is given a set $S \subset V$, and is allowed to start constructing the clustering based on that.

1) Then the remain $V \setminus S$ vertices arrive in an online fashion (in any order, and the means even adversarially), and the algorithm should start making online irrevocable decisions.

In this model, the authors provide the following collection of results:

1) They show that if $S$ is adversarially chosen, then any algorithm should have competitive ratio $\Omega(n)$.

2) If $S$ is a randomly chosen subset of $V$ with size $\epsilon n$ for some $\epsilon \in (0,1)$, then the online extension of an already known algorithm (called Pivot) gives a $O(\frac{1}{\epsilon})$ competitive solution. Moreover, they show that is this case, the ratio $O(\frac{1}{\epsilon})$ is the best possible.

3) They authors also consider a case where an adversary can corrupt the "advice" $S$. Here, the adversary can choose $\alpha n$ elements of $S$, with the rest $(\epsilon-\alpha)n$ ones being again randomly selected. In this scenario, they show that the pivot algorithm achieves a $O(\frac{\epsilon}{(\epsilon-\alpha)^2})$ competitive solution. Finally, the latter result is complemented by a hardness theorem, showing in this scenario the best achievable ratio is $\Omega(\frac{1}{\epsilon - \alpha})$, and hence the given result is almost optimal.

Finally the authors perform a substantial experimental evaluation that validates their theoretical results.

**Limitations And Societal Impact:**

Yes

**Main Review:**

CLARITY:

The paper in general is very well written, and I must admit that I enjoyed reading it. The only suggestion regarding clarity that I have is the following. In the description of semi-online Pivot in page 4, the author should have made it more clear that when a node $v$ arrives in the online phase, if there exists some pivot $u \in S$ such that $\{u,v\} \in E_+$ then $v$ will be placed in the cluster defined by $u$. In the current format, I feel like someone must read between the lines in order to understand this vital point.

SIGNIFICANCE-ORIGINALITY:

To me the main contribution of this work is the introduction of the semi-online model for correlation clustering. From a practical perspective, it seems very reasonable to have some "advice" in the form of a random set $S \subset V$. However, I'm not very convinced about the way adversarial corruption is model in the paper. I understand the concept of letting the adversary choose the set $A$ of size $\alpha n$, but it seems more natural for the random set $R$ to be chosen from $V$ and not from $V \setminus A$. How can this change alter your robust result?

CORRECTNESS-TECHNICAL DETALS:

I went through all the proofs, except the proof of Theorem 4 in Appendix A,  and all arguments appear sound to me. I really liked the proof of Theorem 1, and I think the way the authors broke down the analysis of semi-online pivot is quite elegant.

CONFIDENCE:

I have to admit that I am not really familiar with the literature on correlation clustering (my area of expertise is metric clustering). Nonetheless, from my perspective and given a brief look of related work, I still think that there is merit to this paper and that its contribution is actually significant.


**Time Spent Reviewing:**

6 hours reading the paper + 1 hour thinking about the review + 1 hour writing the review

---

> ### Author Response · Authors · 2021-08-04
> **Clarifying questions of Reviewer uASx**
>
> - However, I'm not very convinced about the way adversarial corruption is model in the paper. I understand the concept of letting the adversary choose the set $A$ of size  $\alpha n$, but it seems more natural for the random set to be chosen from  $V$ and not from  $V \setminus A$. How can this change alter your robust result?
>
> We thank the reviewer for this great insight. This does not significantly change the robustness result and the competitiveness ratio is asymptotically the same. To see why, we first make the observation that when pivoting with the samples, having duplicates does not affect the final clustering at all. Thus we can assume we sample from $V$ instead of $V \setminus A$ first, and then discard all vertices that appear in $A$. This is very close to sampling uniformly from $V \setminus A$ and gives similar theoretical guarantees.

---

### Official Review · Reviewer_PVDb · 2021-07-15

**Rating:** 7
**Confidence:** 3

**Summary:**

The paper considers the online correlation clustering problem in the context of having a sort of a soft start with an offline phase where they get a subset of vertices and the induced subgraph. They provide a modification of the pivot algorithm to carry out the online correlation clustering under the defined model. The algorithm has a $O(1/\epsilon)$ competitive ratio if allowed a $\epsilon n$ size initial offline set. The paper also argues about the robustness of the algorithm even when the $S$ could have adversarial corruptions.

**Ethical Concerns:**

No major ethical concerns as far as I can see.

**Limitations And Societal Impact:**

The paper does not necessarily discuss the limitations. The major limitation that could be noticed was that since the $S$ is derived from $V$ randomly we need to know the set of vertices. But this seems to be a direct artifact of the model.

As the problem itself is a classical problem I don’t think it creates any major negative societal impacts.

**Main Review:**

The paper is well written and the arguments seem sound. They introduce a novel approach to use the established pivot algorithm along with an offline set of vertices to get an algorithm that gives a $O(1/\epsilon)$ competitive ratio, compared to $\Omega(n)$ bounds in the online setting (though they work in slightly different settings). The robustness arguments seem to provide additional support to the pivot algorithm being competitive in this setting. They also manage to provide matching lower bounds as well.

Typos:
140: “then Pivor is a 3-approximation in expectation” => “then Pivot is a 3-approximation in expectation”

Notes:
Please check the citation style instructions and make sure to use \cite and \citet appropriately.
Make algorithms slightly more readable if possible (using algorithm environments or something).

**Time Spent Reviewing:**

2

---

### Official Review · Reviewer_BymN · 2021-07-19

**Rating:** 7
**Confidence:** 3

**Summary:**

The paper studies the standard correlation clustering problem in the *semi-online* model. In the most studied variant of correlation clustering, we are given a complete graph where each edge is either +1 or -1 and the goal is to partition vertices into clusters such that the total number of *disagreements* with the edge signs is minimize. The number of disagreements is the number of negative edges in clusters plus the number of positive edges between clusters. In the offline setting, the problem admits a constant factor approximation, but it is not possible to get a better than $\Omega(n)$-competitive ration in the (adversarial) online setting.

**Limitations And Societal Impact:**

I do not foresee any such issues.

**Main Review:**

This paper investigates the problem under the semi-online model. The intuition is that while the problem is hard under the general online setting, a simple Pivot algorithm gets competitive ratio 3 when the nodes arrive in a random order. To limit the amount of randomness in the sequence, authors proposed to consider the setting in which a randomly selected $\epsilon$-fraction of nodes is given to the algorithm and then the rest of the input arrive in an adversarial order. They proved that with this limited amount of randomness (or advice) the standard Pivot algorithm gets $O(1/\epsilon)$-competitive ratio and it is tight. Further, they considered the robust variant of the semi-online model where the adversary first picks an $\alpha$-fraction of nodes, then a randomly picked $(\epsilon-\alpha)$-fraction of nodes is combined with the adversary’s pick and feed to the algorithm. Even in this case, under some mild condition the Pivot algorithm achieves $1/(\epsilon-\alpha)$-competitive ratio; in other words, the adversary cannot do a significant harm.  Besides, the paper has a nice connection to a recently popular line of research on learning-based algorithms (aka algorithms with predictions). In general, the algorithm and its analysis is clean and nice and I recommend acceptance of the paper.

**Minor Comments**
1.	Line 42. a irrevocable -> an irrevocable
2.	Line 140. Pivor -> Pivot
3.	Line 142. algroithm -> algorithm
4.	Line 304. sparsities -> sparsifies
5.	Mention related work on correlation clustering on random order online setting.
6.	What happens if we change the order of items 1 and 2 in the robust model? Do we know that it becomes significantly harder? What is its status.


**Time Spent Reviewing:**

2

---

> ### Author Response · Authors · 2021-08-04
> **Clarifying questions of Reviewer BymN**
>
> - What happens if we change the order of items 1 and 2 in the robust model? Do we know that it becomes significantly harder? What is its status.
>
> This was asked by the prior reviewer, and we duplicate the response here.
>
> In our model, the order of events is both for technical convenience and tractability. Changing the order of events can result in the problem becoming unsolvable or technically harder, depending on what changes we allow the adversary to make. If the optimal solution is allowed to remove $\alpha n$ nodes from the given samples, the competitive ratio degrades to $\Omega( \alpha n)$. To see this, consider an instance where there is a subgraph of size $\alpha n$ in G and the remaining $(1-\alpha)n$ nodes have negative edges to all other nodes (intuitively the $\alpha n$ nodes are an isolated instance in the larger instance). After the sampling is done, the adversary can choose to eliminate all vertices sampled from this subgraph. The sample set provides no information about this subgraph at all.  Thus, this instance gives the same approximation bounds of $\Omega( \alpha n)$ as in the completely adversarial setting (line 94-95). This illustrates that it is important that the adversary can only add to the random sample and not remove from it.
>
> However, if the adversary can only add $\alpha n$ vertices to the random sample, then the order of events is mainly for technical convenience. One can consider a similar model to ours where first we draw a random sample of size $(\epsilon - \alpha) n$ and then the adversary chooses another $\alpha n$ vertices to add to the sample. The technical difficulty here is that the choice of the adversary depends on the random sample, which does not occur in our model.

---

### Official Review · Reviewer_AUhK · 2021-08-01

**Rating:** 7
**Confidence:** 3

**Summary:**

This paper studies the effectiveness of the Pivot algorithm for the online correlation clustering problem under the mild assumption that a small sample of data is available beforehand. It provides improved guarantees for the algorithm's competitive ratio under the beyond worst-case analysis paradigm, specifically, it shows that the known competitive ratio lower bound can be bypassed using a semi-online algorithm.
A rigorous theoretical analysis is provided, and it is complemented by experimental evaluations.

**Ethical Concerns:**

None.

**Limitations And Societal Impact:**

Some of the limitations are discussed. Assessing societal impact is not applicable.

**Main Review:**

# Originality

Although the algorithm(pivot) that is being analyzed is not new, considering the recent interests in beyond worst-case scenarios, this paper introduces novel insights on how one can design online algorithms for similar problems. In my opinion, the semi-online setting is the correct lens to analyze and design algorithms for this problem and I believe this paper formulates this notion in a clean way.

# Quality

The quality of the writing and the presentation is good in general. The results of this paper are placed well among the prior work and the contributions of this paper are clearly stated.
Following are some of the typos.
- line 22: used in a downstream -> used in downstream
- line 140: fix reference to Ailon et at. [2008] at the end of the sentence
- line 322: compete graphs -> complete graphs

# Clarity

The clarity for the most part is good however, it can be improved in certain places.
-  In the proof of theorem 1, line 267, 268 can the authors explain why the second inequality holds ($\mathbb{E}[T'] \le ...$)? is it because if this is not the case, OPT could have done better?
- In experiments, in temporal advice(lines 370-375), what is the adversarial order when you state "including temporal order"? Can the authors please clarify more how the test data arrive in this case?

Apart from these, I have some further questions.
- In section 4 random sample with adversarial corruptions in line 279, why is the order of events important? What happens for example if a random sample of $\epsilon n$ is first chosen and an adversary changes an arbitrary fraction of size $\alpha n$ from that sample? is there any technical difficulty for choosing this order of events?
- Why is the competitive ratio in theorem 4 "almost" matching the lower bound in theorem 5? what are the difficulties of closing this gap? Can the authors please comment on this?

# Significance

Having some prior knowledge of a small random sample in most cases is feasible and this paper shows how that can be exploited cleverly to develop an algorithm with guarantees much better than what is known from prior lower bounds. This and the fact that one can still hope to perform better with an adversarially corrupted sample up to a certain extent makes this result significant in my opinion.

Overall, I believe this paper is very nicely written and adds nice contributions to the online correlation clustering problem.

**Time Spent Reviewing:**

5

---

> ### Author Response · Authors · 2021-08-04
> **Clarifying questions of Reviewer AUhK**
>
> - In the proof of theorem 1, line 267, 268 can the authors explain why the second inequality holds? is it because if this is not the case, OPT could have done better?
>
> The second inequality bounds the number of bad triangles $T’$ in the remaining graph $G’$ (defined in line 259) after pivoting with the first $\epsilon n$ randomized vertices.  Notice that we have Lemma 2, which bounds the number of bad triangles in a graph using the disagreements in any clustering. The inequality then follows by applying Lemma 2 to the graph $G’$ and the clustering induced by OPT on $G’$, which makes disagreements exactly $E' \cap E^*$ (line 266).
>
> - In experiments, in temporal advice(lines 370-375), what is the adversarial order when you state "including temporal order"? Can the authors please clarify more how the test data arrive in this case?
>
> Here the temporal order is still the same as in previous experiment design (line 337-340). There is a timestamp on each vertex and we let the vertices arrive in order of their timestamps. The only difference is the source of offline advice. In theory and previous experiment designs we assume it comes from the current set of vertices (which might be unknown in advance). Here we show that we can easily obtain advice from historical data that is almost as powerful.
>
> - In section 4 random sample with adversarial corruptions in line 279, why is the order of events important? What happens for example if a random sample of $\epsilon n$ is first chosen and an adversary changes an arbitrary fraction of size $\alpha n$
> from that sample? is there any technical difficulty for choosing this order of events?
>
> In our model, the order of events is both for technical convenience and tractability. Changing the order of events can result in the problem becoming unsolvable or technically harder, depending on what changes we allow the adversary to make. If the optimal solution is allowed to remove $\alpha n$ nodes from the given samples, the competitive ratio degrades to $\Omega( \alpha n)$. To see this, consider an instance where there is a subgraph of size $\alpha n$ in G and the remaining $(1-\alpha)n$ nodes have negative edges to all other nodes (intuitively the $\alpha n$ nodes are an isolated instance in the larger instance). After the sampling is done, the adversary can choose to eliminate all vertices sampled from this subgraph. The sample set provides no information about this subgraph at all.  Thus, this instance gives the same approximation bounds of $\Omega( \alpha n)$ as in the completely adversarial setting (line 94-95). This illustrates that  it is important that the adversary can only add to the random sample and not remove.
>
> However, if the adversary can only add $\alpha n$ vertices to the random sample, then the order of events is mainly for technical convenience. One can consider a similar model to ours where first we draw a random sample of size $(\epsilon - \alpha) n$ and then the adversary chooses another $\alpha n$ vertices to add to the sample. The technical difficulty here is that the choice of the adversary depends on the random sample, which does not occur in our model.
>
>
> - Why is the competitive ratio in theorem 4 "almost" matching the lower bound in theorem 5? what are the difficulties of closing this gap? Can the authors please comment on this?
>
> The competitive ratio matches the lower bound (asymptotically) if $\alpha$ is at most a constant fraction of $\epsilon$ (because in this case $\epsilon - \alpha = \Theta(\epsilon)$.) The gap arises (on the upper-bound side) from our sparsification lemma (lemma 7, line 583 in supplementary material), because we can only bound the expected 'remaining' positive degree by $O(\frac{\epsilon}{(\epsilon - \alpha)^2})$. On the lower-bound side (theorem 5), the gap arises because currently, our proof requires $\epsilon$ and $\epsilon - \alpha$ to be simultaneously small (bounded by some constant), which is not the case is $\epsilon$ is large and $\alpha = (1 - o(1)) \epsilon$ for example. It is conceivable that the gap can be closed by more refined estimates, but for clarity, we have chosen to make the assumption that $\alpha$ is at most a constant fraction of $\epsilon$.

---

> > ### Comment · Reviewer_AUhK · 2021-09-01
> > **Thank you for the clarifications**
> >
> > I thank the authors for the clarifications. I have read the other reviews and the author's responses.
> > I maintain my evaluation.

---

### Decision · Program_Chairs · 2021-09-27

**Decision:**

Accept (Poster)

**Comment:**

The paper considers an online version of correlation clustering and shows that the well-known Pivot algorithm performs well in the setting. The result is not that surprising, but I think this is an important contribution nonetheless and warrants acceptance.